# First Molecular Characterization of Chronic Hepatitis B Carriers in Timbuktu, Mali

**DOI:** 10.3390/diagnostics13030375

**Published:** 2023-01-19

**Authors:** Philip Lawrence, Mawlouda Chabane, Lucie Abrouk, Adrien Thiesson, Diakaridia Berthé, Amadou B. Diarra, Karim Bengaly, Brehima Traoré, Djibril Kassogué, Geoffroy Durand, Catherine Voegele, Florence Le Calvez-Kelm, Nicolas Steenkeste, Pierre Hainaut, Bourema Kouriba, Emmanuelle Gormally

**Affiliations:** 1CONFLUENCE: Sciences et Humanités Confluence (EA 1598), Université Catholique de Lyon (UCLy), 69002 Lyon, France; 2Centre d’Infectiologie Charles Mérieux, Bamako BPE2283, Mali; 3Hôpital Régional de Tombouctou, Timbuktu, Mali; 4Centre National de Transfusion Sanguine (CNTS, National Blood Bank), Bamako BPE1520, Mali; 5International Agency for Research on Cancer, 69008 Lyon, France; 6Fondation Mérieux, 69002 Lyon, France; 7Institute of Advanced Biosciences, INSERM U1209, CNRS UMR 5309, Université Grenoble Alpes, 38700 La Tronche, France

**Keywords:** hepatitis B virus, hepatocellular carcinoma, aflatoxin, *TP53 R249S*, HBV Genotype E, HBV HBx, Timbuktu, Mali

## Abstract

In Mali, hepatocellular carcinoma (HCC) is the third and sixth most common cancer in men and women, respectively. Mali comprises several distinct climato-ecological zones. Most studies to date have been conducted in the sub-Sahelian zone of southern Mali, including the capital city Bamako. In this part of the country, the main risk factors for HCC are chronic hepatitis B virus (HBV) carriage and dietary exposure to aflatoxins, a well-known hepatocarcinogen. Data are scarce for other ecological zones, but our preliminary data from 721 blood donors in the area of Timbuktu, presented in this study, suggest that chronic HBV carriage is also endemic in the northern Saharan zone of Mali. For further study, 29 healthy HBV chronic carrier volunteers were recruited from the blood transfusion center in Timbuktu. Successful viral genotyping in 20 volunteers revealed HBV genotype E in 13 cases and D in 7 cases, suggesting that this geographical and anthropological transition zone may also represent a transition zone between HBV genotypes that dominate sub-Saharan and northern Africa, respectively. Sequencing of circulating cell-free plasma DNA (cfDNA) from donors did not reveal the presence of the *TP53 R249S* mutation in these donors, a marker of dietary exposure to aflatoxins in sub-Saharan Africa. These results suggest that the geo-epidemiological distribution of the risk factors for HCC is not uniform across Mali, but is dependent upon climatic, socioeconomic and anthropological factors that might have an impact on patterns of chronic liver disease and cancer.

## 1. Introduction

Chronic hepatitis B is a public health concern affecting 257 million people worldwide and causing over 800,000 deaths annually from chronic liver disease, cirrhosis and hepatocellular carcinoma (HCC) [1]. Chronic hepatitis B virus (HBV) infection is a major risk factor for HCC, accounting for at least 50% of cases of HCC worldwide and 80% of HCC caused by viruses [2]. Rates of chronic HBV carriage vary between 0.2% and 22.38% of the general population, with the highest rates in sub-Saharan Africa and eastern Asia, where the infection is endemic (>8%). In these areas, transmission is frequently vertical or perinatal, resulting in HBV chronic carriage in 90% of infected infants [1]. In other parts of the world, infection by HBV occurs mostly through sexual or parenteral routes later in life, which leads to chronic HBV carriage in less than 10% of cases.

HBV is classified into ten genotypes, including eight established (A to H) and two putative genotypes (I and J), for which approximately 40 sub-genotypes have been identified [3,4]. HBV genotypes display distinct geographical and ethnic distribution and there is evidence that they also differ by their pathogenic properties, including their risk of persistence as chronic infection and their capacity to induce precursor liver disease or HCC [4]. In Africa, the most common HBV genotypes are A (mainly in central and southern Africa), E (western Africa) and D (northern Africa). The risk of HCC is compounded by a multiplicative interaction between HBV and aflatoxin, a class of mycotoxins produced by a fungi of *Aspergillus* sp. that grows on staples, such as cereals and groundnuts, and thus contaminates the diet in regions with hot, humid climates. Specifically, the metabolization of aflatoxin B1 in hepatocytes generates bulky adducts on guanines, including the third position of codon 249 of the *TP53* gene, leading to a G-to-T transversion (AGG to AGT, arginine (R) to serine (S)) [5]. The resulting mutant allele predominates in cases of HCC occurring in the context of joint HBV carriage and dietary exposure to aflatoxins. Traces of mutant DNA at codon 249 are detectable in circulating cell-free DNA (cfDNA) extracted from the plasma or serum of HCC patients, as well as of individuals exposed to dietary aflatoxins [5,6,7,8,9,10,11]. This suggests that the presence of the circulating mutation could be interpreted as a biomarker of exposure to aflatoxins in subjects with no evidence of liver disease or cancer in countries where the mycotoxin is known to infest the food chain [5,7,11].

Mali belongs to the Economic Community of West African States (ECOWAS). Broadly, the country encompasses two broad and major climato-ecological zones, including the so-called Sudano-Sahelian (savanna) zone in the southern part of the country (containing the capital city Bamako) and the Sahara zone in the northern part. Mali is an ethno-linguistic mosaic where over 60 ethnic groups and 20 languages coexist (OECD, http://www.oecd.org/swac/maps/, accessed on 9 December 2022). These populations are differently distributed between the two zones, with a majority of Mande (Bambara and Malinké) in the south, while the north is mostly populated by Tuareg, Moor and Songhay ethnic groups. In Mali, HCC is the 3rd and 6th most common cancer in men and women, respectively [12] (Global Cancer Observatory, https://gco.iarc.fr/, accessed on 9 December 2022). Studies on the epidemiology of HCC in Mali are limited to the southern zone, where, as in the rest of west Africa, most cases occur in a context of chronic carriage of HBV genotype E and exposure to aflatoxins through the diet. In a previous study on 1466 volunteers from the Bamako district, we found that 18.8% of donors were chronic HBV carriers [13], consistent with previous reports focusing on specific population groups (pregnant women, blood donors and students) [14,15,16,17,18,19]. HBV genotype E was present in 91.1% of the carriers. Genotype A or D were found in only 8.9% of carriers, co-infected or not with genotype E. Biomarker-based studies (Fibrotests, Actitest) revealed that over one-third of carriers had markers of liver fibrosis and/or active viral infection, which would, in principle, make them eligible for anti-HBV therapy. Traces of *TP53 R249S* mutant cfDNA were found in 60% of chronic carriers, suggesting low levels of ongoing exposure to dietary aflatoxins [13]. North of Mali, in northern Africa, chronic HBV carriage occurs at rates that vary between 1.3% (Morocco) and 4% (Egypt) [20]. Data for Algeria, which shares a border with Mali, are sparse, but studies have shown a prevalence of HBV carriage between 2 and 7% [21,22]. The dominant genotypes are D and A, detected in 86.5% and 11.8% of HBV carriers from Algeria, respectively [23]. Of note, the majority of HCC cases in north Africa develop in a context of chronic hepatitis C virus (HCV) infection [24].

Amongst chronic HBV carriers, a further biomarker for progression to liver disease, including HCC, involves the presence of mutations or truncations in viral DNA sequences, specifically involving the viral core gene promoter and/or the partially overlapping HBx gene [25,26,27]. For the viral HBx gene, deleted or truncated 3′ fragments are thought to represent mutations in crucial elements for viral replication and may therefore represent replication-defective persistent genomes within host cells. As such, the detection of HBV gene fragments can be considered as a marker of integration of viral DNA in hepatocytes and for the risk of liver disease [26,28,29]. Additionally, for this gene, several recurrent point mutations have been associated with an elevated risk of cirrhosis or HCC when compared with carriers without such mutations [25,30,31,32].

To date, there is no epidemiological or molecular data on HBV chronic carriage in the Saharan region of Mali. In this study, we have carried out a serological and molecular analysis of HBV infection in volunteers recruited at the Timbuktu regional hospital. We show that, as for southern Mali, chronic HBV carriage is endemic in this population (11%). Viral genotyping revealed that this geographical and ethnic transition zone between north and sub-Saharan Africa may also represent a transition zone for HBV genotypes E and D. Contrary to the southern zone of Mali, traces of *TP53 R249S* mutants are undetectable in volunteers from Timbuktu, suggesting that the mutagenic load caused by dietary aflatoxins is low in the northern zone. Analysis of full or partial viral HBx genes from cfDNA revealed the presence of the truncated form of the viral gene, indicative of chronic HBV carriage and a risk for progression to liver disease. These results warrant further studies on risk factors and patterns of chronic liver disease and cancer in these populations.

## 2. Materials and Methods

### 2.1. Subjects and Samples

Thirty HBV chronic carriers were recruited among blood donors at the regional hospital of Timbuktu in 2017 after giving their informed consent. The samples were first tested in Timbuktu for the presence of AgHBs by a rapid test (DETERMINE™HBsAg 2, Abbott Rapid Diagnostics SAS, Evry, France) when the subjects came to donate blood. AgHBs-positive blood donors were enrolled for the study. A total of 10 mL of blood was collected from each volunteer and centrifuged at 4000 rpm for 5 min to collect plasma and serum. The plasma and serum were processed within half a day (between 15 min and 5 h) and stored at −20 °C until they were transported to the Centre d’Infectiologie Charles Mérieux in Bamako (Mali), where they were again stored at −20 °C. The study protocol was approved by the ethical committee of the FMPOS (Faculté de Médecine Pharmacie et d’Odonto-Stomatologie of Bamako, decision N°2016/80/CE/FMPOS, 27 June 2016) according to national regulations in Mali. Prior to inclusion into the study, the subjects received individual information and a written informed consent form was obtained.

As a reference for phylogenetic analysis, thirty samples from HBV chronic carriers recruited at the Centre National de Transfusion Sanguine (CNTS, National Blood Bank), Bamako, Mali in an ongoing study were included. The HBV chronic carriers were selected according to their AgHBe status (15 AgHBe positive and 15 AgHBe negative subjects). PCR and genotype analyses were carried out in the same way as described below for samples from Timbuktu.

All gene sequences were submitted to GenBank under the following accession numbers—Timbuktu: HBS genes ON806988–ON807008, HBX genes ON806949–ON806960; Bamako: HBS genes ON807009–ON807038, HBX genes ON806961–ON806987. Appropriate reference sequences were retrieved from the HBV database (https://hbvdb.lyon.inserm.fr/HBVdb/, accessed on 22 September 2020) and included in the phylogenetic analysis as indicated.

### 2.2. HBV Viral Load

Extraction of the HBV virus DNA from EDTA plasma samples was performed on the high-performance Arrow/LIAISON IXT^®^ (DiaSorin Ireland Ltd., Dublin, Ireland) machine following the manufacturer’s recommendations. The viral load was determined by quantitative PCR using the Generic HBV viral load test from BIOCENTRIC (Bandol, France) and CFX96™ Real-Time System BIO RAD^®^ thermocycler (Hercules, CA, USA) following the manufacturer’s specifications.

### 2.3. Serology—AgHBs, AcHBs, AcHBe, AgHBc

The plasma samples were tested at the Centre d’Infectiologie Charles Mérieux (Bamako, Mali) by ELISA for AgHBs (Surase B-96 TMB, General Biologicals Corporation, Taiwan), IgG anti-HBs (Antisurase B-96 TMB, General Biologicals Corporation, Hsinchu County, Taiwan), AgHBe and IgG anti-HBe (EASE BN-96 TMB, General Biologicals Corporation, Hsinchu County, Taiwan) according to the manufacturer’s recommendations.

### 2.4. HBV Genotyping

HBV genotyping was performed by sequencing the viral HBV surface gene (HBS) in cfDNA extracted from the plasma (500 μL) using the QIAamp^®^ Circulating Nucleic Acid Kit (Qiagen, Courtaboeuf, France). A semi-nested PCR was used to amplify the entire S gene, as described previously [11]. Briefly, the first reaction was performed using 1 μL of DNA with primers S_HBVpol1 (5′-CCTGCTGGTGGCTCCAGTTCA-3′) and S_HBVporv2 (5′-AAAGCCCAAAAGACCCACAAT-3′) in a 20 µL reaction to amplify an 800 base pair fragment. The second step used 1 μL of the first reaction product and primers S_HBV123s (5′-TCGAGGATTGGGGACCCTG-3′) and S_HBVporv2. For each sample, at least one sequencing reaction was performed for the forward and reverse sequence from two independent PCR products. Direct sequencing was performed on amplified fragments using the primers S_HBV123s, S_HBVporv2 and S_HBV778r (5′-GAGGTATAAAGGGACTCAAG-3′). The obtained sequences were blasted, and the genotype was determined using the HBV genotyping tool from the Hepatitis B Virus database (HBVdb, https://hbvdb.ibcp.fr/HBVdb/HBVdbGenotype, accessed on 8 September 2020). To validate the genotype, phylogenetic trees were also determined using S gene sequences and the Software MegaX [33]. Determination of the serotype/sub-genotype of HBV using protein sequences is described elsewhere [34,35].

It was possible to amplify the 800 bp fragment of the HBS gene from seven of the 30 Timbuktu DNA samples. To amplify the remaining samples, primers were designed to amplify three smaller fragments, including the sequences essential for genotyping and PCR conditions were optimized. A smaller region of the S gene was consequently amplified in three overlapping PCR fragments. The three PCR reactions used 1 μL of DNA with primer pairs HBS-2S (5′-TCGTGGTGGACTTCTCTCAA-3′) and HBS-2R (5′-TTAGAGGACAAACGGGCAAC-3′) that amplify a fragment of 229 bp, HBS-3S (5′-ATCCTGCTGCTATGCCTCAT-3′) and HBS-3R (5′-AAACTGAGCCAGGAGAAACG-3′) that amplify a fragment of 269 bp and HBS-4S (5′-aACCTTCGGACGGAAATTG-3′) and HBS-4R (5′-CCCAAAGACAAAAGAAAATTGG-3′) that amplify a fragment of 249 bp. The primers were used at a concentration of 0.2 µmol.L^−1^ with the Taq polymerase from the 2x Purple Master Mix (Ozyme, Saint-Cyr-l’École, France) following the manufacturer’s recommendations. For the PCR reaction, the DNA was initially denatured for 5 min at 95 °C. Then cycles including a denaturation at 95 °C for 30 s, a hybridization of primers at 62 °C for primer pairs HBS-2S/2R and HBS-3S/3R or 50 °C for primer pair HS-4S/4R for 30 s and an extension of DNA at 72 °C for 1 min were repeated 50 times. A final extension was performed at 72 °C for 5 min. Sense and anti-sense strands were sequenced with forward and reverse primers for each DNA fragment. For each sample, the sequences were obtained from at least two different PCR reactions. The sequence of the three fragments were then assembled and the obtained sequence was analyzed bioinformatically as described above.

### 2.5. Analysis of the HBV HBx Gene Status

The detection of truncated versus complete HBx gene sequences was performed by PCR as previously described [29,36] with modified methods [27]. Briefly, 1 µL of cfDNA was amplified by PCR to produce four overlapping DNA fragments from HBx of either 139, 192, 334 or 425 bp. The 425 bp DNA fragment encompasses the entire HBx sequence and the three shorter DNA fragments correspond to fragments initiated at the 5′-end of HBx and covering progressive lengths of its sequence. Amplification of all four fragments signals the presence of a complete HBx, whereas amplification of one or several shorter fragments signals the presence of 3′-truncated HBx. The same forward primer X1F was used in each reaction (5′-GGGACGTCCTTTGTCTACGT-3′) together with one of four reverse primers: X1R (5′-GGGAGACCGCGTAAAGAGAG-3′), X2R (5′-GTGCAGAGGTGAAGCGAAGT-3′), X3R (5′-CCCAACTCCTCCCAGTCTTT-3′) or X4R (5′-GGAGAGGTGAAAAAGTTGCA-3′). The HBx status of each sample was confirmed by a second PCR and for each sample, the longest fragment was sequenced with forward and reverse primers from two distinct PCR reactions. The optimization of all protocols was performed using plasma samples obtained from the biobank ICAReB of the Pasteur Institut (Paris, France). An HBV genotype E HBx gene consensus protein sequence was obtained by alignment of 380 available, non-duplicate sequences from the HBV database (https://hbvdb.lyon.inserm.fr/HBVdb/, accessed on 22 September 2020).

### 2.6. Phylogenetic Analysis of Viral HBS Genes

Phylogenetic trees were constructed using the maximum likelihood method and Tamura–Nei model [37]. The initial trees for the heuristic search were obtained automatically by applying the Neighbor-Join and BIONJ algorithms to a matrix of pairwise distances estimated using the Tamura–Nei model, and then selecting the topology with superior log likelihood value. The trees are drawn to scale, with branch lengths measured in the number of substitutions per site. Evolutionary analyses were conducted in MEGA X [33,38].

### 2.7. TP53 R249S Amplification, Library Construction and Deep Sequencing with Ion Torrent Proton

As the size of cfDNA fragments in cancer patients was recently reported to follow a narrow-range, unimodal distribution reaching a peak at 166 bp [39], the primers were designed to amplify 83 bp of the partial exon 7, covering codons 237 to 261 (hg19: ch17: 7,577,489–ch17: 7,577,571). The forward and reverse primer sequences were 5′-TGTGTAACAGTTCCTGCAT-3′ and 5′-GGCTCCTGACCTGGAGTCTT-3′, respectively. PCR amplification was performed using 5 ng of cfDNA, 5x AccuStart Buffer, 200 nM of forward and reverse primers and 0.04 U/mL of AccuStart HiFi Taq Polymerase (Quanta BioSciences, Beverly, MA, USA) with the following conditions: 2 min at 94 °C, 40 cycles of 30 s at 94 °C, 30 s at 58 °C and 40 s at 72 °C and a final elongation of 5 min at 72 °C. Approximately 20% of the PCR products were quantified by a QubitTM dsDNA HS Assay Kit and (Invitrogen, Thermo Fisher Scientific, Waltham, MA, USA) and Qubit^®^ 2.0 fluorometer and 20 ng of the PCR product were purified with Serapure magnetic beads at a final concentration of 2.5x and 28% of isopropanol. The library preparation was performed using the NEB Next^®^ Fast DNA Library Prep Set for Ion Torrent™ kit (New England Biolabs, Ipswich, MA, USA) with some modifications, where each volume of reagent was reduced by a factor of 4. Briefly, 12.5 μL of the 20 μL purified products were end-repaired in 15 μL, and added to 8.6 μL of the ligation reaction mix, 0.7 μL of the Ion P1 Adapter and 0.7 μL of each Ion Barcode for the ligation step. The barcoded products were purified using the Serapure magnetic beads at a final concentration of 1.8x, amplified in 25 μL and quantified using the Qubit quantification system. A total of 40 ng of the amplified barcoded products were pooled into a single tube and the cleanup and size selection of the pooled libraries (~180 bp) was performed in a 2% agarose gel and MinElute Gel Extraction Kit (Qiagen, Courtaboeuf, France). The pool of purified barcoded libraries was quantified using the Qubit quantification system and the assessment of the library quality (molarity and size analysis) was performed using the Agilent^®^ High Sensitivity DNA Kit and the Agilent Technologies 2100 BioanalyzerTM (Agilent Technologies, Santa Clara, CA, USA). Emulsion amplification was performed on the Ion OneTouch 2 system (Thermo Fisher Scientific) using 7 μL of 100 pM library and the Ion PI Hi-Q OT2 200 Kit (Thermo Fisher Scientific), according to the manufacturer’s protocol. Quality control of Ion PI Ion Sphere Particles was performed using the Qubit 2.0 fluorometer, as described in the protocol. The sequencing reaction was performed on an Ion Proton System (Thermo Fisher Scientific) using Life Technologies’ Ion PI™ Chip Kit v3 and Ion PGM™ Hi-Q™ Sequencing Kit (Thermo Fisher Scientific), according to the manufacturer’s instructions. The library preparation and sequencing conditions were adapted from previous protocols [40,41,42].

### 2.8. Bioinformatics and Statistical Analyses

This study used Needlestack, a developed variant caller algorithm suitable for the detection of low-abundance mutations [40,41,42,43] (https://github.com/IARCbioinfo/needlestack, accessed on 9 October 2018). This approach is based on the inclusion of sequencing data of a sufficient number of samples to robustly estimate the sequencing error rates at each position considered and for each possible base change. The reads were mapped to the human whole genome and the BAM files were generated by the Ion Torrent Proton server using default parameters. Reads with a base quality below thirteen at the position TP53 c.747G were excluded from subsequent analysis. For TP53 R249S, the sequencing error was modeled using a negative binomial regression [44] to avoid bias in parameters estimation due to the potential presence of genetic variants detected as being outliers from this error model. For each sample, a p-value for being a variant (outlier from the regression) was calculated that was further transformed into q-values to account for multiple testing. The q-values are reported in Phred scale Q = −10 log10(q-value) and a threshold of Q > 20 was used to call the variants.

## 3. Results

### 3.1. Characteristics of the Hepatitis B Carriers for Timbuktu, Mali

The blood donor center in Timbuktu is located in the regional hospital of Timbuktu city. In 2016, the serum of 721 blood donors was tested routinely with a rapid diagnostic test for the presence of HBV surface antigen (AgHBs). Eighty blood donors tested positive for AgHBs. This would indicate that within the tested population, 11% were HBV chronic carriers. Of those 80 positive for AgHBs, only 46 had a telephone number in the records of the hospital. Between January and March 2017, these 46 AgHBs-positive blood donors were called back, but only 15 agreed to participate in the study. The AgHBs status of those 15 volunteers was again tested and confirmed chronic carriage of the virus. During the same period, a further 15 blood donors who tested positive for the AgHBs were enrolled in the study. For these donors, chronic carriage of the virus was not tested by a second AgHBs assay on a different blood sample collected six months after the first. However, given the fact that the volunteers were healthy and that most HBV transmission in this part of the world occurs in early childhood, it is reasonable to assume that these 15 donors also represent chronic carriers. The characteristics of the 30 volunteers are described in Table 1. They were mostly men (80%), on average 32.8 years old, with tertiary sector occupations (83%) and living within Timbuktu city.

### 3.2. Serological and Viral Load Characteristics of Chronic Carriers in Timbuktu, Mali

The AgHBs antigen was detected twice by a rapid test in the blood of the 15 blood donor volunteers at an interval of six months and only once for 15 blood donors, as mentioned above. The results were confirmed on a VIDAS HBs Ultra (bioMérieux, Marcy-l’Étoile, France). The AgHBs-positive status for one volunteer was not confirmed (Table 2). The subject was found negative for all other serological markers. All the volunteers were HBV AgHBe-negative and anti-AgHBe (AcHBe)-positive except for the individual who was not confirmed to be AgHBs-positive. If the consistently AgHBs-negative subject is not considered, 93% of the Timbuktu HBV chronic carriers were found to be AcHBc-negative.

More than half (66.7%) of the HBV chronic carriers had an undetectable viral load, while 30% had a viral load above 20,000 UI/mL, characteristic of active HBV chronic infection. These results are similar to our previous study on chronic carriers in Bamako, Mali [13], for which we found that 33% of HBV chronic carriers had viral loads indicative of active infection.

### 3.3. Genotype and HBx Characteristics of HBV Chronic Carriers in Timbuktu and Bamako

As a reference group for further molecular and phylogenetic analysis in this study, 30 samples from donors recruited at the Centre National de Transfusion Sanguine (CNTS, National Blood Bank), Bamako, Mali, in a parallel study were included. The characteristics of these 30 volunteers are described in Table 1 and Table 2. All volunteers were men, with an average age of 29.9 years old. The 30 volunteers from Bamako were all confirmed HBV chronic carriers by two AgHBs positive results from two samples taken at an interval of six months (50% AgHBe positive, 50% AgHBe negative).

The HBV genotype for the 30 HBV chronic carriers from Timbuktu and the 30 HBV chronic carriers from Bamako was determined using two different approaches. With the first approach, we were able to amplify cfDNA from the 30 chronic carriers from Bamako and seven HBV chronic carriers from Timbuktu using a method already described [11]. For the remaining samples from Timbuktu, it was assumed that the samples that tested negative with this PCR contained shorter cfDNA fragments that could not be amplified by the primers previously described. Therefore, three smaller PCR amplicons spanning the HBV S gene were used. Using this method, the 30 Timbuktu samples were tested and an additional 13 sequences were amplified for this group of donors. In total, 20 of the 30 donors from Timbuktu and all of the chronic carriers from Bamako could be amplified for genotyping. The results of the genotyping are shown in Table 2. The HBV genotypes were further confirmed by comparison to known HBV sequences obtained from the HBV database (https://hbvdb.lyon.inserm.fr/HBVdb/, accessed on 22 September 2020) by phylogenetic analysis (Figure 1).

In agreement with the results of our previous study [13], in Bamako we found 28 HBV genotype E (93.3%) and two HBV genotype A (6.7%), while for the 20 samples from the Timbuktu volunteers that could be amplified, the HBV genotype distribution was different, with 13 HBV genotype E (65%) and seven HBV genotype D (35%).

In a more detailed phylogenetic analysis, as expected, the HBV genotype E from Bamako and Timbuktu were found to cluster generally with HBV genotype E sequences from sub-Saharan Africa (Figure 2), confirming the low genetic diversity of this genotype [3,45]. The seven HBV genotype D from Timbuktu were found to cluster together, closest to an HBV D7 subtype from Tunisia. The two HBV genotype A from Bamako clustered with HBV A3/A4 genotypes of suspected African origins, notably the Gabonese Republic (Figure 2).

Given the well-established links between HBx sequence mutations and the course and outcome of liver disease and HCC in chronically infected patients [25,26,27], we also decided to amplify cfDNA HBx gene sequences from donor serum samples obtained for this study by nested-PCR. Full or partial HBx gene fragments could be amplified from 27/30 samples from Bamako and from 15/29 samples from Timbuktu (Table 2). For samples from Bamako, 13 of the samples (48.1%) were found to contain 3′-truncated forms of HBx and, as expected, these truncated forms mostly corresponded to an AgHBe-negative status for these chronic carriers. For Timbuktu, only two of the 15 PCR positive samples (13.3%) contained full-length HBx, whilst the remaining 13 serum samples (86.7%) contained shorter 3′-truncated forms (Figure 3).

A further analysis of the HBx gene sequences for genotype E from Timbuktu and Bamako confirmed that the gene sequence is globally conserved between the two regions for the genotype E strains when compared to a subtype E consensus sequence for the viral gene (Figure 3). The amino acid change rates for the HBx protein between the two cohorts were between 2 and 3% when compared with a genotype E consensus sequence (2.4% for Bamako sequences and 3% for Timbuktu sequences).

For the viral HBx gene, several recurrent point mutations have been suggested to be indicative of an elevated risk of cirrhosis or HCC when compared with carriers without these mutations [25,30,31,32]. In this sense, further analysis of the HBx genotype E sequences obtained from cfDNA was performed. In agreement with previous studies, commonly identified mutation sites were observed, including at amino acid positions 88 (9/33 sequences), 127 (3/17 sequences) and the double mutation at positions 130/131 (6/15 sequences) (Figure 3). In addition, novel mutation hotspots were observed at positions 22 (S22G) (9/35 sequences) and 144 (A144V/P) (6/13 sequences) when compared to a subtype E consensus sequence. No differences were observed between the mutation frequency or patterns between sequences obtained from carriers from Timbuktu or from Bamako.

### 3.4. Exposure to Alfatoxin in Bamako and Timbuktu

To measure the exposure to aflatoxin in the Timbuktu and Bamako volunteers, the TP53 mutant *R249S* load was quantified by ultra-deep next generation sequencing (NGS) and a customized algorithm specifically developed to detect low-abundance mutations. The median sequencing read depth of the 59 analyzed samples was 49,608X. Of the 30 Timbuktu volunteers, 29 could be tested for the presence of the TP53 R249S mutation in cfDNA. Twenty-eight samples tested negative for the presence of the mutant, while one sample failed to give interpretable results. As a comparison, the 30 HBV chronic carriers from Bamako were also tested for the presence of the TP53 R249S mutant in cfDNA. One sample failed and three samples (10.3%) contained low detectable quantities of TP53 R249S mutants (allelic fractions of 1.16%, 0.51% and 0.04%), indicative of exposure to aflatoxin B1 (data not shown).

## 4. Discussion

We present here for the first time a molecular characterization of HBV genotypes circulating in northern Mali, specifically in the region of Timbuktu. From the HBV chronic carriers diagnosed in the blood transfusion center of Timbuktu, a prevalence of infection with HBV could be roughly estimated at 11%, characteristic of endemic regions and similar to studies in Bamako, which found HBV chronic carriage incidences between 10 and 18% [13,14,15,16,17,18,19].

The region of Timbuktu is a geographic and ethnic transition zone between sub-Saharan Africa, where a majority of HBV genotype E circulates, and north Africa, where a majority of HBV genotype D are diagnosed. For the 30 Timbuktu HBV chronic carriers who volunteered, 13 were found to harbor an HBV genotype E (65%), seven a genotype D (35%), nine could not be determined and, for one subject, chronic carriage of HBV could not be confirmed. In comparison, in the 30 HBV chronic carriers from Bamako, the majority were found to carry a genotype E (93%) and two a genotype A (7%), which is in general agreement with our previous study [13]. The two subjects from Bamako with HBV genotype A were found to cluster with genotypes A3 and A4 after phylogenetic analysis of the viral HBS gene. Given that the genetic distribution of HBV subtypes in Africa is increasingly shown to be more diverse and complex than previously thought, notably for subtype A genotypes [46], the fact that different HBV genotypes can display different molecular and functional characteristics that can influence clinical manifestations [47,48,49], and the lack of detailed knowledge for this area of the Sahara/northern Mali, it will be of interest to further investigate the nature of these Malian HBV sequences with whole genome sequencing and analysis where sample quality allows.

Viral loads in Timbuktu indicated active HBV replication in 33.3% of volunteers (>20,000 UI/mL), in agreement with our previous study for volunteers in Bamako (32.2% >10,000 copies/mL) [13]. In this study, the volunteers in Bamako were specifically selected as a reference group for molecular and phylogenetic analysis using their AgHBe status and high viral load. These results would indicate that over 30% of the HBV carriers in Timbuktu should benefit from an antiviral HBV treatment. This, together with the endemic HBV chronic infection (estimated at 11% of the population), would advocate for more generalized screening of the Timbuktu population for HBV chronic carriage in order to propose a treatment and follow-up to prevent liver diseases, cirrhosis or HCC.

It is also interesting to note that a high proportion of subjects (13/15 HBx amplified sequences) from Timbuktu presented truncated forms of the viral HBx gene in cfDNA. The presence of such gene fragments in cfDNA is a common marker of integration of viral DNA in hepatocytes and of progression to liver disease including HCC [26,28,29]. Indeed, deleted or truncated 3′ HBx fragments are thought to affect crucial elements for viral replication, including the promoter for the viral core gene or encapsidation signals and may therefore represent replication-defective persistent genomes within host cells. Indeed, deletions in the HBx gene have been more commonly associated with tumor-derived sequences compared to non-tumor-derived sequences [28].

For a number of samples from Timbuktu (14/29), we were unable to amplify even short 5′ HBx sequence fragments from the cfDNA despite the use of different primers and conditions. This would suggest further possible truncations or even deletions in this viral gene, although it should also be considered that this difference may also reflect a lower sample quality and nucleic acid degradation linked to sampling, storage and transport difficulties for the samples from this region of Mali. Indeed, for six subjects out of 29, it was not possible to amplify either HBs or HBx. Overall, given the relatively high proportion of subjects for which truncated HBx fragments could be amplified in this study and the potential association of such gene fragments in cfDNA with disease progression and prognosis, it would again seem to be of vital interest to continue studies in this region of Mali for which data are currently lacking concerning cases of liver disease or HCC. The mutation analysis of the HBx sequence fragments from the cfDNA also confirmed the presence of several common mutation sites that have been suggested to be associated with an increased incidence of liver disease and/or HCC [25,30,31,50]. Of note, the well-documented mutations at amino acid positions 88 and the double mutation at positions 130/131 (nucleotides 1762/1764) were present in 27% and 40% of subjects presenting fragments at these positions, respectively. Novel mutational hotspots were observed at positions 22 (S22G) (22% of subjects) and 144 (A144V/P) (46%) when compared to a subtype E consensus sequence.

*TP53 R249S* mutations in cfDNA were analyzed in the Timbuktu and Bamako volunteers by NGS. No mutations were detected in the Timbuktu sera and 10.3% of the Bamako (3/29) sera had detectable low-level mutant DNA fragments. This difference between Bamako and Timbuktu might be explained by differences in diet and, therefore, in exposure to dietary aflatoxin in the two regions of Mali. However, our sample size is too small to completely exclude dietary exposure to aflatoxin in the region of Timbuktu. Moreover, all volunteers from Timbuktu were recruited in the same season (January to March). For reference, a study in The Gambia by Villar et al. (2011) found levels of *TP53 R249S* mutations in cfDNA five times lower between October and March compared to levels observed between April and July [11]. It will be of interest to increase the number of volunteers in future studies and to extend the recruitment period to further investigate the seasonality of exposure to aflatoxin in this region of Mali.

Of note, our previous study in Bamako used SOMA (Short Oligonucleotide Mass Analysis) and had found that in 2009, 60% of the volunteers had detectable levels of *TP53 R249S* mutations in cfDNA [13]. While we could not compare the sensitivity and specificity of the two technologies used for detection (SOMA vs. NGS), the ultra-deep sequencing method combined with the Needlestack ultra-sensitive variant caller [43] used in this and a previous study [10] enables the detection of very low-level (as low as 0.04%) *TP53 R249S* mutations in cfDNA. In this respect, it is interesting to note that for Bamako, levels of aflatoxin exposure seem to be lower in the present study than in our previous one. Such a difference in incidence of exposure in aflatoxin B1 in Bamako could be explained by changes in diet in the district of Bamako between 2009, the time of recruitment for our first study, and 2017 (the time of recruitment for the present study), although we do not have any data confirming such changes. The difference in aflatoxin B1 exposure between the two studies might also be explained by differences in population recruitment for the two studies. Based on this study alone, it is difficult to conclude that exposure to aflatoxin has decreased for Bamako in the period between 2009 and 2017. Given the reported correlation between aflatoxin exposure, the appearance of *TP53 R249S* mutations and incidence of HCC, further studies will be needed to confirm a decrease in aflatoxin exposure levels in Bamako considering current aflatoxin reduction policies in Mali and throughout the world (https://www.ifpri.org/publication/aflacontrol-project-reducing-spread-aflatoxins-mali, accessed on 9 December 2022) [51].

To conclude, we report here for the first time a molecular characterization of HBV chronic infection in the region of Timbuktu, a region that represents a transition zone between northern and sub-Saharan Africa. Our data suggest that chronic HBV carriage is also endemic in this region of Mali (11%). Strikingly, 30% of subjects would appear to display active HBV replication, highlighting the need for population level diagnosis, patient follow-up and treatment in the region. As expected for this transition region between the Sahara Desert and the Sahel, the HBV genotypes found in chronic carriers are different than those found in Bamako, with 65% of genotype E (93% in Bamako) and 35% of genotype D, the predominant genotype in northern Africa. However, given the lack of data for liver diseases and HCC incidence in Timbuktu, it is difficult to evaluate the impact that such a difference in genotype distribution might have for these diseases and their clinical management. These preliminary results warrant further studies to confirm the incidence of the two genotypes and their impact on liver diseases and HCC appearance and for the development and clinical management of HBV infection in the region of Timbuktu.

## Figures and Tables

**Figure 1 diagnostics-13-00375-f001:**
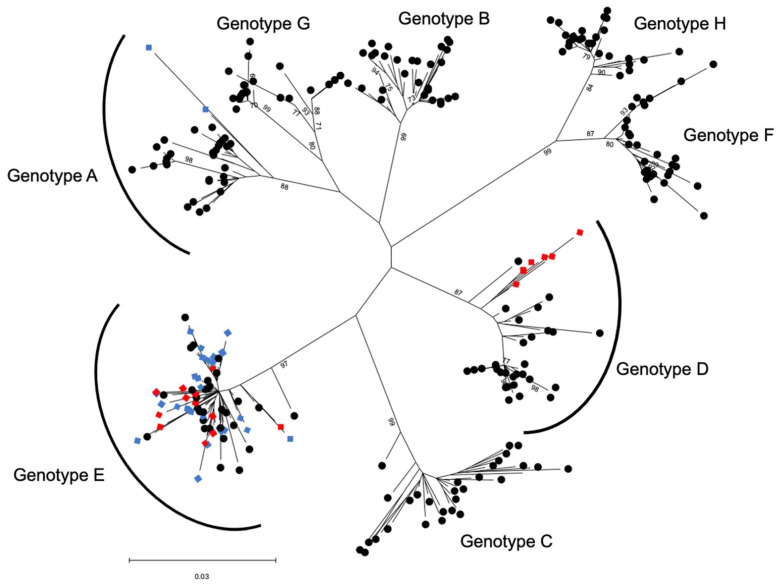
Phylogenetic analysis of HBV sequences based on 969 nt of the viral HBS gene compared to reference sequences for each viral genotype A–H. The phylogenetic tree was constructed using the maximum likelihood method and the Tamura–Nei model [37]. The tree with the highest log likelihood (−8124.86) is shown. This analysis involved 284 nucleotide sequences with a total of 969 positions in the final dataset. The points shown with blue diamonds represent samples collected in Bamako (Mali) and the points shown with red diamonds represent samples from Timbuktu (Mali). The black circles represent reference sequences.

**Figure 2 diagnostics-13-00375-f002:**
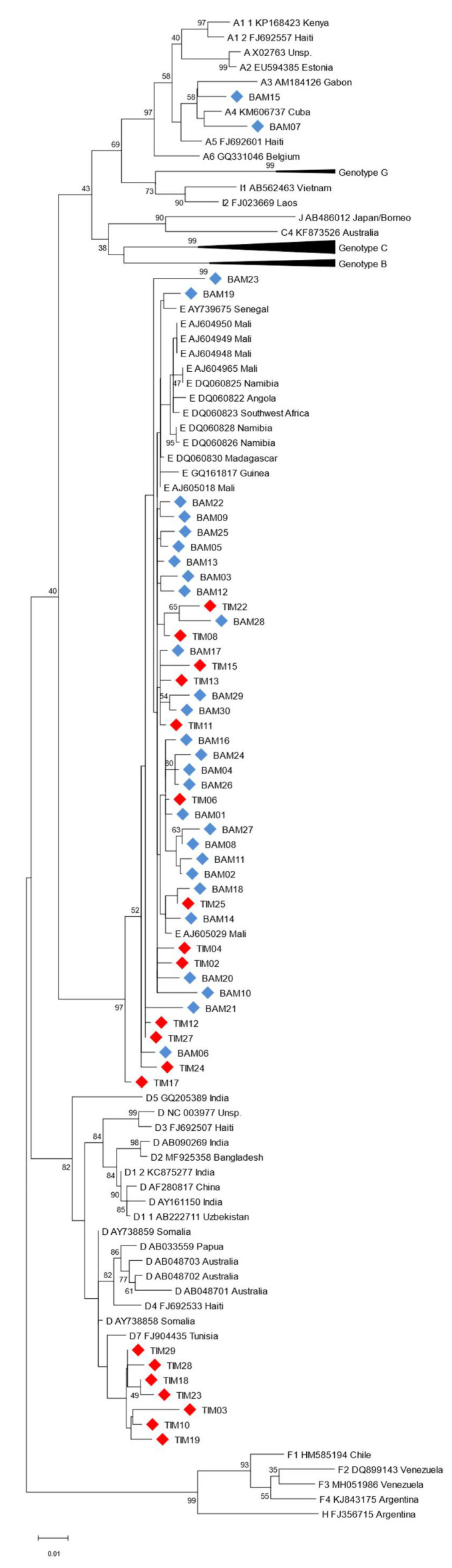
Phylogenetic analysis of HBV sequences based on 964 nt of the viral HBS gene compared to HBV reference sequences. The phylogenetic tree was constructed using the maximum likelihood method and the Tamura–Nei model [37]. The tree with the highest log likelihood (−9422.64) is shown. The percentage of trees in which the associated taxa clustered together is shown next to the branches. This analysis involved 118 nucleotide sequences with a total of 964 positions in the final dataset. Reference sequences are shown as Genotype, GenBank accession number and country of origin (where specified). The points shown with blue diamonds and labeled BAM represent samples collected in Bamako (Mali) and the points shown with red diamonds and labeled TIM represent samples collected in Timbuktu (Mali).

**Figure 3 diagnostics-13-00375-f003:**
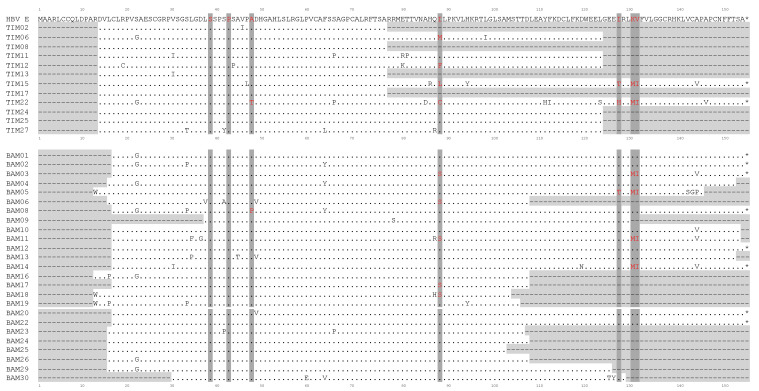
Comparison of HBV HBx protein sequences (154 aa) for 37 chronic carriers presenting with HBV genotype E in this study. HBV E refers to a consensus aa sequence obtained by comparison of 380 available non-duplicate sequences from the HBV database (https://hbvdb.lyon.inserm.fr/HBVdb/, accessed on 22 September 2020). The points indicate unchanged aa compared to the reference sequence. Areas shaded in light gray (dotted lines) indicate non-amplifiable regions. * indicates a stop codon. Areas shaded in dark grey (aa in red) show point mutation positions commonly associated with HCC in the literature. TIM, Timbuktu (Mali); BAM, Bamako (Mali).

**Table 1 diagnostics-13-00375-t001:** Characteristics of the 60 volunteers recruited at the regional hospital of Timbuktu (29 AgHBs-positive subjects) and the national blood transfusion center of Bamako (30 AgHBs-positive subjects).

Parameters	Participants in Timbuktu	Percentage (%)	Participants in Bamako	Percentage (%)
**Age group**				
18–29	15	50	13	43
30–39	3	10	14	47
40–49	10	33.3	3	10
50–59	2	6.7	0	0
**Gender**				
Male	24	80	30	100
Female	5	20	0	0
**Occupation**				
Employee (private or civil)	10	33.3	UN	UN
Student	9	30	UN	UN
Trader	6	20	UN	UN
Farmer	4	13.3	UN	UN
Housewife	1	3.3	UN	UN

UN: unknown.

**Table 2 diagnostics-13-00375-t002:** Serology, viral load, genotyping and HBx status for AgHBs-positive volunteers from Timbuktu and Bamako (Mali).

	Timbuktu		Bamako	
Parameters	Number (Total 30 Samples)	Percentage (%)	Number (Total 30 Samples)	Percentage (%)
**Serology**				
AgHBs Timbuktu				
Positive	29	96.7	NR	NR
Negative	0	0	NR	NR
Unknown	1	3.3	NR	NR
AgHBs Centre d’Infectiologie Charles Mérieux				
Positive	29	96.7	30	100
Negative	1	3.3	0	0
AcHBs				
Positive	2	6.7	3	10
Negative	28	93.3	27	90
AgHBe				
Positive	0	0	15	50
Negative	30	100	15	50
AcHBe				
Positive	29	96.7	NT	NT
Negative	1	3.3	NT	NT
AcHBc				
Positive	3	10	2	6.7
Negative	27	90	28	93.3
**Viral load**				
Undetectable	20	66.7	0	0
Between 2000 and 20,000 UI/ml	1	3.3	1	3.3
>20,000 UI/ml	9	30	29	96.7
**Genotyping of HBV**				
Genotype E	13	44.8	28	93.3
Genotype D	7	24.1	0	0
Genotype A	0	0	2	6.7
Could not be determined	9	31.1	0	0
**HBx status**				
Full length HBx	2	3.4	14	43.3
Truncated HBx	13	51.7	13	50
No amplification product (failure of PCR/deletion of HBx)	14	41.4	3	6.7

NR: not relevant for these samples; NT: not tested.

## Data Availability

All gene sequences obtained and analyzed for this study were submitted to GenBank under the following accession numbers—Timbuktu: HBS genes ON806988–ON807008, HBX genes ON806949–ON806960; Bamako: HBS genes ON807009–ON807038, HBX genes ON806961–ON806987.

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
