# Peer review of "First Molecular Characterization of Chronic Hepatitis B Carriers in Timbuktu, Mali"

_diagnostics, 2023, doi:10.3390/diagnostics13030375_

Round 1
Reviewer 1 Report
This is a very carefully written manuscript describing advanced classification of HBV isolates. The phylogenetic interpretations are neither over- nor understated and break new ground.
Minor comment:
Section 3.4
The link between alfatoxin and mutation rates is made to TP53 and is highly relevant diagnostically as it is detected in cfDNA samples. However, How do the collected viral and somatic genomic data compare in this respect? What is the authors' opinion as to a link between alfatoxin exposure and HBV mutation/evolution rates? Is that worth discussing?
Might it be that the geographical link to alphatoxin exposure is not explicitly addressed by the collected data set? If so, this link could be reserved for the discussion rather than the results section?
Author Response
We would like to thank reviewer one for their careful reading of the manuscript and their stimulating questions.
We would like to specify that we do not have access to genomic DNA and therefore, we have not compared TP53 mutations in cfDNA and genomic DNA for these volunteers. We have only analysed TP53 codon 249 mutations in cfDNA, which is thought to represent DNA coming from hepatocytes and is often used as a marker of Aflatoxin exposure.
In their second comment, the reviewer raises the question of HBV mutation following exposure to Aflatoxin. There is no clear established direct link in the literature that Aflatoxin mutates HBV. However, there is data in the literature suggesting that HBV genotype E has co-evolved with hepatocytes carrying the TP53 aflatoxin mutants giving these cells evolutionary advantages. However, this study is primarily a description of HBV carriage and aflatoxin exposure in the populations of Tombouctou and Bamako. In this respect, discussing co-evolution of HBV genotype E with TP53 codon 7 mutants might be out of the scope of this study and would not necessarily help in the discussion of the results. It remains however, a very interesting biological question that would require further in vitro studies using molecular and cellular models.
We completely agree that the TP53 S249R data does not prove that there is exposure to Aflatoxin in Bamako but not in Timbuktu. In the results section, we simply describe the results and in the discussion, we comment the result but we feel that we remain very cautious as to the interpretation of a difference in exposure between the two regions, particularly given the relative low number of samples and the collection of samples in a very limited period of the year. There is currently no data from Timbuktu on the level of aflatoxin in cereal crops upon which to base further speculation.
Reviewer 2 Report
The current manuscript has demonstrated extensive study for the regional detection of prevalence of HBV carrier in Timbuktu region.
1. However, one major concern is the limited sample size (n=29) of the current study for genotyping chronic HBV carrier. The sample size is much less than their previous study in which n=87 (PMID: 25886382, or ref 13) or other HBV carrier studies (Ref 14-19). Moreover, half of the blood donors succeeding for HBV genotyping were only assumed to be chronic HBV carrier (Line 287-294). Whether the 29 subjects can be representative to chronic HBV carriers in the whole Timbuktu region is in question. Indeed, among the 29 subjects, only 20 got successful in PCR (Line 333). 9 of the remaining failed. The reason for PCR failure is multiple, either technical error or true unknown genotype. That is to say only 20 subjects of Timbuktu region were truly tested for genotyping. Moreover, how to define the subjects recruited to Timbuktu regional hospital are local to Timbuktu and be representative to the place? It will be great to see if authors could address these point.
2. In addition,Line 442-453, the R249S and S249R were confusing and require revision.
3. For figure 1, labels are overlapped and difficult to visualize. It will be great to have a clearer version. Moreover, a rooted tree will be better to show the potential origin and neighbourhood of Genotype E and D from Timbuktu.
4. Statistic analysis (e.g. T test/confidence interval test) is required when authors want to address differences or similarities between Timbuktu and Bamako.
Author Response
We would like to thank reviewer two for their careful reading of the manuscript and their stimulating questions.
- We completely agree that the sample size is very small. Since 2013, Timbuktu is a very difficult area to reach and to collaborate with scientifically. However, there is a real wish from our collaborators to continue research in this part of the world so as to improve patient care. In 2017, we were able to have a trained collaborator in Timbuktu only for 2 months. Since 2017, the region is too unstable to take the responsibility to send a scientist there.
In effect, we cannot exclude that the lack of results for HBs for 9 subjects are due to technical problems. We have repeated the PCR multiple times and developed a new assay to screen and reassemble shorter fragments. We favor the hypothesis that the DNA samples from Timbuktu were partially degraded because of transport conditions.
Regarding the origin of volunteers, their address was recorded in the questionnaire when they were enrolled. We have carefully mapped those addresses which were all in Timbuktu city. There always remains the possibility that the volunteer gives an address of a family member in Timbuktu.
As discussed in the manuscript we cannot be sure that 15 volunteers are chronic carriers of HBV. However, in this part of the world most infections occur in early childhood. We keep in mind that our hypothesis is that Timbuktu is a transitional zone also for the etiology of HBV infection and that in this region we cannot exclude that compared to Bamako there are more infection during adulthood especially with the genotype D.
2. This is an error and has been modified in the text (line 446) to indicate R249S.
3. The authors agree with remarks concerning figure 1 and the labels have been removed for clarity. The legend for figure 1 has been slightly modified as a result to reflect these changes. Figure 1 shows an unrooted maximum-likelihood tree to illustrate and confirm genotype distribution between Timbuktu and Bamako. The rooted version of the tree is provided as Figure 2 to show more information for potential relations and origins between sequences as discussed in the manuscript (lines 363-369 & 470-478), although the low genetic diversity typical of genotype E makes it difficult to infer much more information in this regard.
4. We had considered this approach to analyze the mutation data between Bamako and Timbuktu. However, our data is non-parametric (different sample sizes (Bamako n=25 sequences and Timbuktu n=12 sequences) and does not have a normal distribution (Shapiro-Wilk test p=0.000067). We feel that using another test than a t-test would not bring any value to the analysis of the data since the sample size is small. To avoid any confusion in this sense, we have modified the text to remove any suggestion that our interpretation included any statistical analysis. We remain at your disposal if you require any further analysis in this respect.